# Identification of Crown of Thorns Starfish (COTS) using Convolutional Neural Network (CNN) and attention model

**Maleika Heenaye- Mamode Khan** [1]☯*, **Anjana Makoonlall**[1]☯, **Nadeem Nazurally**[2]☯, **Zahra Mungloo- Dilmohamud**[3]☯

1 Department of Software and Information Systems, Faculty of Information, Communication and Digital Technologies, University of Mauritius, Reduit, Mauritius, 2 Department of Agricultural and Food Science, Faculty of Agriculture, University of Mauritius, Reduit, Mauritius, 3 Department of Digital Technologies, Faculty of Information, Communication and Digital Technologies, University of Mauritius, Reduit, Mauritius

☯ These authors contributed equally to this work.
* m.mamodekhan@uom.ac.mu

**Data Availability Statement:** The study's minimal underlying data set can be accessed as follows: Crown-of-thorns starfish: https://www.kaggle.com/

## Abstract

Coral reefs play important roles in the marine ecosystem, from providing shelter to aquatic lives to being a source of income to others. However, they are in danger from outbreaks of species like the Crown of Thorns Starfish (COTS) and the widespread coral bleaching from rising sea temperatures. The identification of COTS for detecting outbreaks is a challenging task and is often done through snorkelling and diving activities with limited range, where strong currents result in poor image capture, damage of capturing equipment, and are of high risks. This paper proposes a novel approach for the automatic detection of COTS based Convolutional Neural Network (CNN) with an enhanced attention module. Different pre-trained CNN models, namely, VGG19 and MobileNetV2 have been applied to our dataset with the aim of detecting and classifying COTS using transfer learning. The architecture of the pre-trained models was optimised using ADAM optimisers and an accuracy of 87.1% was achieved for VGG19 and 80.2% for the MobileNetV2. The attention model was developed and added to the CNN to determine which features in the starfish were influencing the classification. The enhanced model attained an accuracy of 92.6% while explaining the causal features in COTS. The mean average precision of the enhanced VGG-19 with the addition of the attention model was 95% showing an increase of 2% compared to only the enhanced VGG-19 model.

## Introduction

Coral reefs are often referred to as rainforests in terms of their huge biodiversity. Like rainforests, coral reefs in the ocean harbour a wide variety of marine organisms. Coral reefs play a vital part in the marine ecosystem. Other than supporting marine species, coral reefs help in controlling carbon dioxide in the ocean, in protecting coastal areas from erosion and natural calamities and in attracting tourists, thus bringing money to a country [1]. Unfortunately,

datasets/antonsibilev/crownofthorns-starfish
Crown_of_thorns_starfish: https://www.kaggle.com/datasets/hugonaya/crown-of-thorns-starfish
Non- Crown_of_thorns_starfish: https://www.kaggle.com/datasets/sonainjamil/bleached-corals-detection

**Funding:** The author(s) received no specific funding for this work.

**Competing interests:** The authors have declared that no competing interests exist

these coral reefs are facing various natural and human-related threats; including overfishing, global warming and different types of pollution. In Mauritius, it has been observed in certain lagoons that more than 75% of corals have been lost due to a rise in temperature [2]. The sustainability of coral reefs is being jeopardised by drastic temperature fluctuations, resulting in mass coral bleaching and infectious disease epidemics. In the Maldives, the comprehensive response of 191 coral species at depths ranging from 3 to 30 metres revealed a significant decrease in bleaching effects with depth [3]. It can therefore be concluded that temperature and depth have a strong correlation with coral bleaching. Another factor in the degradation of the coral reefs is the outbreak of the *Acanthaster*, commonly known as the Crown of Thorns Starfish (COTS). COTS are a vital part of the ecology on healthy coral reefs when present in normal quantities and controlled by its natural predators. An outbreak of COTS implies that 30 or more adult starfish are present per hectare on reefs [4]. COTS starfish are notorious for wreaking havoc on coral reef ecosystems. This is largely due to the fact that during periodic population breakouts, local densities of COTS can grow from very low densities (1 starfish per hectare) to extremely high densities (>1000 starfish per hectare). Furthermore, COTS are one of the most powerful and effective predators of scleractinian corals. Adult COTS may destroy whole corals, including rather large colonies, whereas most other individual coral-feeding animals (e.g., Chaetodon butterflyfishes and Drupella snails) produce primarily localised lesions or tissue loss. High COTS concentrations will, thus, result in quick and broad short- to long-term coral loss [5]. These outbreaks may be the consequence of overfishing of the crown-of-thorns starfish's main predator, the giant triton, or they can be a natural occurrence [4].

There are two main techniques that are usually adopted to detect COTS. One of them is the recording of videos for a section of the reef and then counting the COTS while reviewing the videos. The other technique is known as the Manta Tow Survey, where a person counts the number of COTS while being towed behind a boat. This recorded number is then extrapolated to determine an approximation on the distribution over several square kilometres. These methods are time consuming and error prone. Furthermore, some COTS populations are underestimated in certain regions when using these techniques [6].

With the recent advances in the field of computer vision and artificial intelligence, there is a growing interest in deploying automated systems dealing with images. To date, there are very few applications which have been developed to automatically detect COTS. The rapid advancement of deep learning is fuelling the community's interest in applying these techniques to develop applications for the COTS detection. CNN, which is a deep learning algorithm, has brought satisfaction to the field of computer vision and has achieved good performance. However, developing a CNN from scratch requires a lot of training data and high-performance graphical processing units (GPUs). In addition, there is a need for the constant adjustment of parameters to deal with overfitting or underfitting models. Another upcoming challenge with deep learning is the lack of transparency in the decision-making process, that is, the identification of the causal parameter.

This paper proposes an innovative approach to detect and classify starfish and aims to determine the features influencing the decision-making process. The rest of this paper is structured as follows. Section 2 describes the state of the art on COTS detection and the background study. Section 3 provides details on the architecture of the system and the models that have been implemented in this work. Section 4 presents and discusses the results obtained. Lastly, Section 5 concludes the paper.

The main contributions of the paper are as follows:

1. Application of transfer learning through the architecture of the pre-trained VGG19 (Visual Geometry Group) and MobileNetV2 models.

2. Enhancement of the CNN models and development of an attention module.

3. Creation of a dataset of local COTS images and thorough testing of the models on the local dataset

4. Development of the attention model and its application on the dataset and an analysis of the features that influence the classification process.

## Related works

The monitoring of coral-eating COTS on a large scale is a challenging task. Multiple attempts to develop automated applications to detect COTS have been made [7]. To build such systems, appropriate datasets are required. There are only a few datasets available online. A dataset of 3157 images has been developed by Dayoub et al., [8] with both COTS and non-COTS instances. Pooloo et al., [9] created a dataset using images publicly available online. The dataset of 134 images was divided into training, validation and testing and in the training dataset, data augmentation was applied. One challenge encountered by researchers is the quality of the images that are being captured underwater [8].

Usually after image capture, it is vital to perform image pre-processing on the dataset before they are used in model training. The two most famous pre-processing methods are Median Filter and Gaussian Filter. According to Singh [10], one of the most fundamental image-smoothing filters is the median filter. It is a non-linear filter that finds the median of surrounding pixels to remove black-and-white noise from an image. Median Filter helps in the preservation of edges. On the other hand, Gaussian Filter is used for blurring an image. It is based on the image's standard deviation and assumes that the mean is zero [11]. It is more effective to remove noise from underwater images.

Object detection is another important aspect of image processing. Object detection that is both efficient and accurate has been a key issue in the progress of computer vision systems [12]. The YOLO (You Only Look Once) technique is a novel emerging method for detecting multiple objects in an image by drawing bounding boxes around them [13]. YOLO is easy to use and can provide real-time object prediction. Pooloo et al., [8] used Efficient-D0 on their custom dataset for the detection of COTS. EfficientDet is an object detection model that employs many optimizations and backbone changes, including the usage of a Weighted Bi-directional Feature Pyramid Network (BiFPN) and a compound scaling strategy that equally scales the resolution, depth, and breadth of all backbones, feature networks, and box/class prediction networks at the same time [13]. Another important concept in object detection is segmentation. Segmentation is the process of dividing a digital image into several segments. The goal of segmentation is to simplify and/or change an image's representation in order to make it more understandable and simpler to study [14]. Techniques that can be used for segmentation are thresholding, edge detection and watershed segmentation. Feature extraction is a formal procedure that identifies the important shape information included in a pattern in order to facilitate pattern classification. In image processing, feature extraction is a sort of dimensionality reduction. The purpose of feature extraction is to extract the most significant information from the original data and represent it in a lower-dimensional space [15]. Local Binary Pattern (LBP) and Histogram of Oriented Gradient (HOG) are the popular machine learning methods used in feature extraction.

In a prior work conducted in Abbasi et al., [16], the authors offered a cooperative dynamic task assignment framework for a specific class of autonomous underwater vehicles (AUVs), used to combat the COTS epidemic on Australia's Great Barrier Reef. The goal of eradicating

COTS clusters utilising the injection system of COTSbot, AUVs transformed the challenge of monitoring and managing COTS into a restricted task assignment problem. A probabilistic map of the operating environment was created, which included seabed terrain, COTS clusters, and coasts. The COTSbot AUVs were then cooperatively injected into the greatest achievable COTS in a given mission time using a novel heuristic method termed Heuristic Fleet Cooperation (HFC). Extensive simulated testing and quantitative performance assessments were carried out to establish the efficacy and long-term viability of the proposed cooperative labour assignment algorithm in eliminating COTS in the Great Barrier Reef.

A vision-based COTS identification and tracking system based on underwater film pictures and a Random Forest Classifier (RFC) was employed in this study by [8]. It used the RFC in a particle filter detector and tracker to monitor COTS with a moving camera, weighting the particles using the RFC's predicted class probability and sparse optical flow estimates for the filter's prediction phase. Using a robotic arm that swings a camera at varying speeds and heights over a range of real-size images of COTS in a reef habitat, the system is experimentally validated in a realistic laboratory scenario.

However, deep learning models are becoming more popular in solving computer vision problems Despite the outstanding results that have emanated from the various deep learning solutions, the issue of computational power remains a big issue. Thus, instead of developing models from scratch, pre-trained models can be adopted. MobileNet is one of the pre-trained models adopted for image classification. Depthwise separable convolutions are used by MobileNet. When compared to the network with ordinary convolutions of the same depth in the nets, it greatly reduces the number of parameters. As a consequence, lightweight deep neural networks are created with MobileNet [17]. It can therefore be used on mobile devices as well as browsers. Another popular pre-trained model is the VGGNet. There are more layers with smaller kernels in VGGNet that help to increase non-linearity. It has increased accuracy and speed [18]. However, pre-trained models have not been explored to detect COTS. In a work conducted by Pooloo et al. [9], a novel reef monitoring strategy based on machine Learning algorithms such as Naïve Bayes, Decision Tree, K-Nearest Neighbour, Support Vector Machine, Random Forest, and XGBoost were presented to automatically categorise corals into varied bleaching severities by training on prior bleaching episodes. SMOTE and optimization methods such as Grid Search and Particle Swarm Optimisation were used to strengthen the experiment. Furthermore, Deep Learning was applied to detect COTS in underwater photos using a specifically trained EfficientDet-D0, resulting in an 81% accurate detection rate. These revolutionary technologies are intended to help marine biologists safeguard reef habitats. Table 1 summarises some work conducted in the identification or classification of COTS.

From the review of the various work conducted so far, it is noticed there are very few works that have applied deep learning on the detection of starfishes, thus, motivating further research in this area.

Inspired from the work conducted by researchers in similar fields of study and in the area of computer vision, we will develop an automated COTS detection application using deep learning, while addressing the challenges.

## Materials and methods

A deep learning model is developed and trained on the COTS datasets. The proposed system allows the user to upload a test image to determine whether it is an image of COTS or not.Fig 1 illustrates the high-level architecture of the system.

**Table 1. Overall review of literature.**

| Paper | Dataset and Techniques used | Performance | Limitations |
|---|---|---|---|
| [16] | Heuristic Fleet Cooperation (HFC) algorithm | Overall accuracy is not provided | Clustering algorithm to group similar objects<br>Limited to only one specific feature |
| [8] | Own dataset of 3157 images<br>Random Forests Classifier and a Particle filter | Precision: 0.98<br>F1-score: 0.98<br>Recall: 0.98 | Unable to discriminate<br>individual COTS when in mass aggregation |
| [9] | 134 different images of COTS from the www<br>Efficient-D0 was adopted | Accuracy:81% | Very small dataset was used<br>Hyperparameter tuning for deep learning was not mentioned. Thus, no mention of model overfitting or underfitting |

## Datasets

There are very few COTS datasets available for the research community. In this work, we have used three online datasets available on Kaggle and have also captured our own images. The first dataset consists of 429 COTS images and is available at [19]. The second dataset contains only 38 COTS images and can be accessed at [20]. The third dataset, which consists of 238 non-COTS images, is available at [21]. Images from all the three datasets were used to obtain a diversity of data. In addition to the above-mentioned images, we have also captured some data by taking photos of COTS in the lagoon around the island of Mauritius. Thus, a total of 506 COTS images and 238 non-COTS images were obtained. Fig 2 provides details of the dataset as well as sample images. As mentioned previously, this is a new area where labelled images are scarce. To create variability and flexibility of our model, data augmentation, which artificially creates training images, was applied. It is also observed that the number of images for the COTS and non-COTS are imbalanced. Therefore, data augmentation will help in resolving the class imbalance issue. Data augmentation is a technique used in deep models to enlarge the dataset in order to alleviate the problem of limited data size. There are several popular data augmentation techniques like flipping, cropping, scaling, rotation, interpolation, translation, and noise insertion. However, not all of them are applicable in different circumstances. In this work, rotation, flipping and distortion were adopted. Eventually, a dataset of 3455 images were created, out of which 1800 were COTS and 1655 were non-COTS.

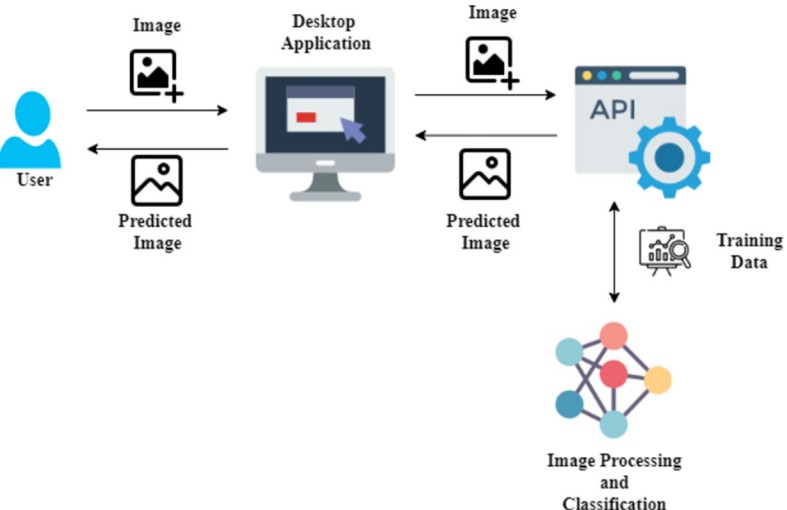

**Fig 1. High-level architecture of the model.**

| Dataset | Samples |
|---|---|
| Crown-of-thorns starfish [19] |  |
| Crown_of_thorns_starfish [20] |  |
| Non- Crown_of_thorns_starfish [21] |  |
| Own captured images |  |

**Fig 2. COTS and non-COTS datasets.**

## Components of the proposed model

Fig 3 represents the main system components of the proposed solution. The modules are image pre-processing, object detection, feature extraction and COTS detection.

In the image pre-processing phase, the COTS images are enhanced and prepared for the next phase. First the starfish is being detected and subtracted from the background images. The features are then extracted and only the relevant ones are selected. The last phase consists of the classification of the image into COTS and Non-COTS.

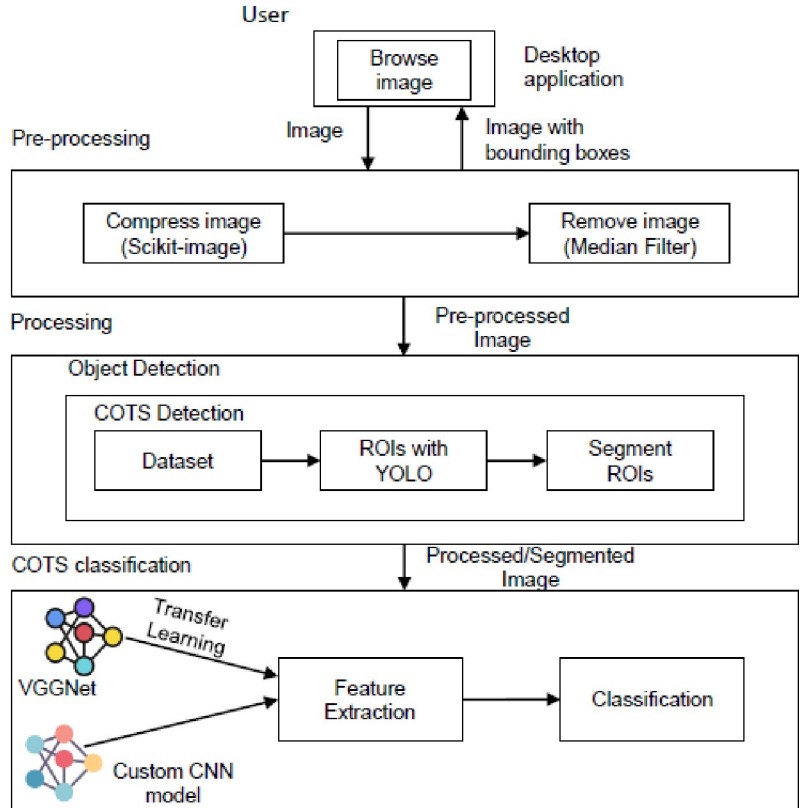

**Fig 3. System components.**

## Noise removal and Image enhancement

The quality of images captured have an effect on the detection phase. Underwater images have more challenges as red wavelengths of light get absorbed faster compared to the blue wavelengths. Clement et al., [6] reported that colour segmentation is not a reliable technique as the colours of COTS vary on dependent factors such as age, location and altitude at which the COTS image was taken. On the other side, COTS do have distinct properties and thus, the texture can be used as a potential feature. Image pre-processing has been used to improve the visual look of the image and to transform the image to a format more suitable for human or machine interpretation [22]. Median filter, Wiener filter, Histogram Equalisation among others are mature techniques usually preferred in the pre-processing phase. However, the domain and types of images captured are the determinant factor which influence the adoption of a particular filter. After a deep analysis and several experiments, the Median filter and the Gaussian filter have been used. In fact, according to Singh [10], one of the most fundamental image-smoothing filters is the Median filter. It is a non-linear filter that finds the median of surrounding pixels to remove black-and-white noise from an image. On the other side, Median filter has been used as it is efficient in noise removal and in salt and pepper noise removal.

## Extraction of features using deep learning through VGG16 and MobileNetV2

Despite the advances of deep learning in various domains, effort to detect COTS using this technique is limited. Instead, research was more focussed on the application of machine

learning. Deep learning is an efficient approach that is composed of multiple layers that learn to represent data with many levels of abstraction [23]. It enables the discovery of complex architectures in high dimensional data by using back propagation algorithms. The deep neural network learns the illustrative and differential features in a systematic way [24, 25]. The architecture of the CNN consists of three types of layers namely the convolutional layers, the pooling layers and the fully connected layers. The image's pixel values will be stored in the input layer. The output of neurons related to particular portions of the input will be determined by the convolutional layer, which will calculate the scalar product between their weights and the region connected to the input volume. The rectified linear unit (ReLU) is performed to apply an activation function like sigmoid to the output of the previous layer's activation. The pooling layer will then just execute down sampling along the input's spatial dimensions, lowering the number of parameters inside that activation even further. When the flattened matrix from the pooling layer is given as an input, a fully connected layer arises, which classifies and labels the pictures [26]. Fig 4 illustrates an image processed using CNN with the different layers as explained in [27].

The formulation of a CNN model is time consuming and computationally expensive. It required a lot of training data for better performance. In addition, high-performance graphical processing units (GPUs) are required for the training of CNN for faster processing [28]. CNN also requires the constant adjustments of parameters to converge to the solution and to overcome overfitting. CNN pre-trained models are those that have already been trained on benchmarked datasets and have learned to extract robust features from images. Thus, the adoption of pre-trained models is highly beneficial.

Despite the promising results of deep learning, the latter has not yet been explored in the detection of COTS. There are several pre-trained models like VGG16, VGG19, ResNet, GoogleLeNet among others that have been trained on benchmarked datasets and have produced remarkable results. In this work VGG19 and MobileNetV2 have been adopted for the deployment of the automated COTS detection system. Instead of using a single Convolution layer with a big kernel size, the VGG network introduced the idea of grouping many convolution layers with smaller kernel sizes. As a result, there were fewer features at the output, and adding three RELU layers rather than one increased the number of learning instances. On the other hand, MobileNet-v2 is a convolutional neural network that is 53 layers deep, which has been trained with over one million images from the ImageNet dataset. Previously machine learning techniques were popular in feature extraction and selection. Machine learning refers to the use of algorithms by computers to learn from data and carry out tasks automatically without explicit programming. In contrast, deep learning employs a powerful set of algorithms that are designed after the human brain. This makes it possible to process unstructured data, including text, photos, and documents. In this case, deep learning is able to extract complex features from the COTS images.

## Application of VGG19 model

Fig 5 illustrates the architecture of transfer learning using VGG19. The VGG19 consists of 16 Convolution Layers, 3 Fully Connected Layers, 5 MaxPool Layers and 1 SoftMax Layer [29] (Lagunas and Garces, 2018). This model has been extensively trained on the ImageNet datasets which consists of millions of images. In our work, weights were initially assigned and updated, while using this architecture. Compared to other models, VGG19 has a deeper CNN architecture with more layers and thus, the reason for its application for our COTS system. However, VGG19 contains 138 million computational parameters. To deal with this high volume of parameters in this deep network, we have utilised a smaller filter of 3x3 as advised in Mateen

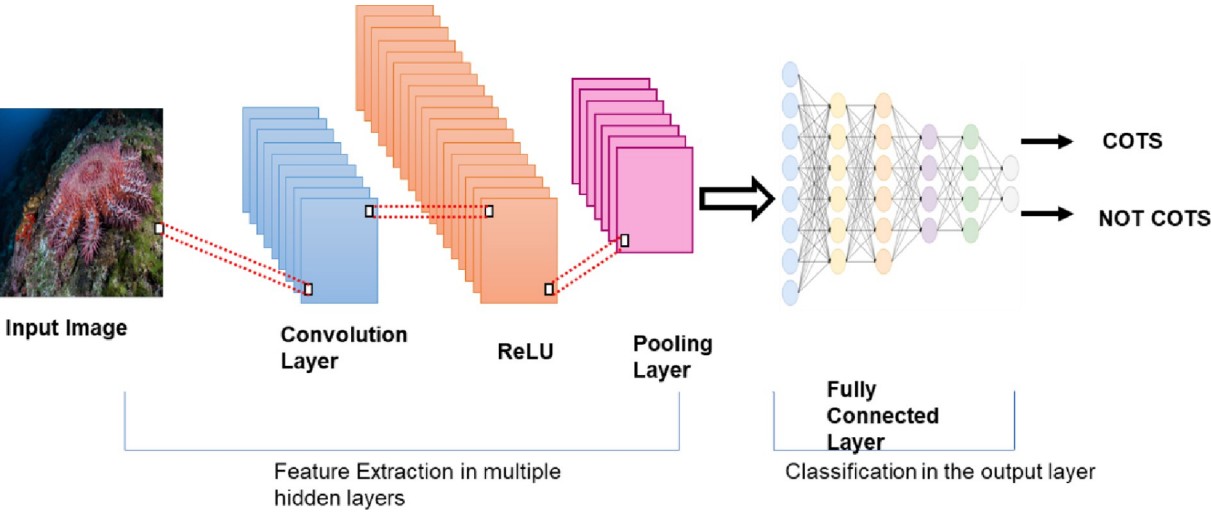

**Fig 4. Architecture of Convolutional Neural Network (CNN).**

et al. [30]. In our work, we have used the last fully connected layer to extract the relevant COTS features. The Softmax function is utilised for the output function of the last layer.

## Application of MobileNetV2 model

To analyse the performance of the pre-trained models, we have also used MobileNetV2. The latter network architecture is made up of 19 unique basic blocks known as bottleneck residual blocks. MobileNetV2 is known to be an architecture that reduces network cost and size. The MobileNetV2 is an enhanced version of MobileNetV1 that solves the problems related to nonlinearities in narrow layers, and therefore caters for the bottlenecks that may crop up [31]. Fig 6 presents the architecture of MobileNetV2. We have opted for MobileNetV2 since it also incorporates depthwise separable filters and has a 1x1 convolution for each input

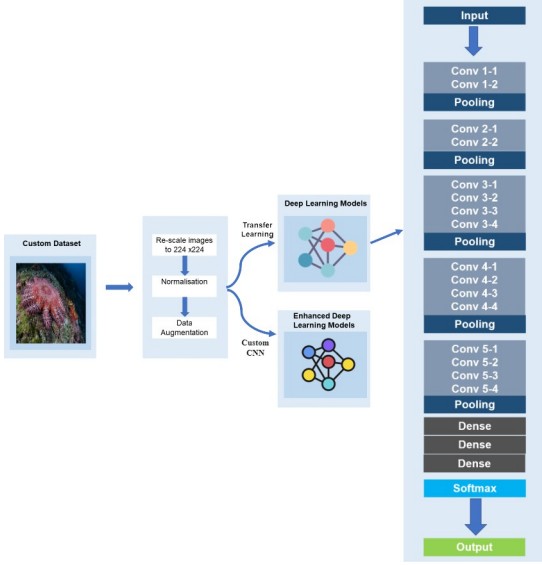

**Fig 5. System architecture with VGG19 layers.**

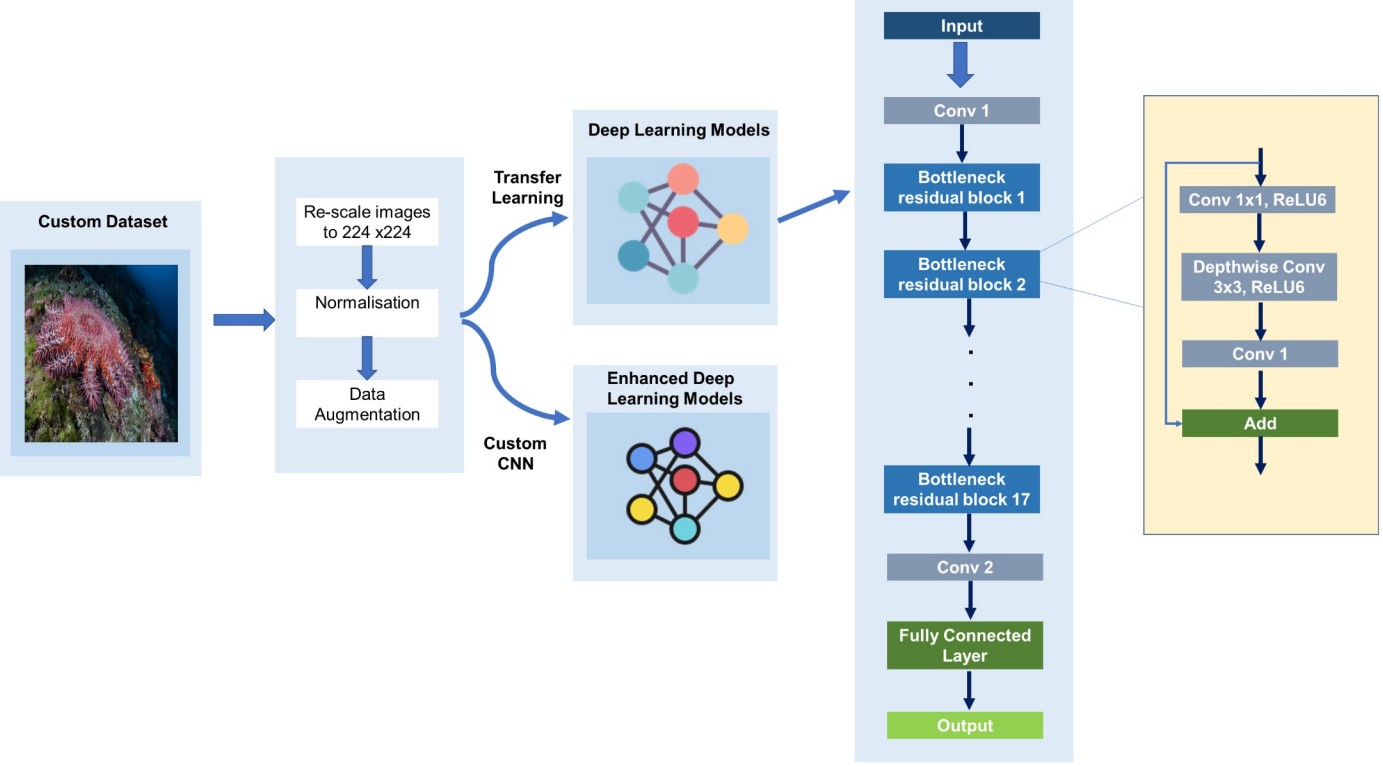

**Fig 6. System architecture with MobileNetV2 layers.**

layer. The depthwise filters examine each input and then separate it into different layers, thus, reducing the speed and cost of the model. In addition, it uses the batchNorm, which supports a high learning rate. This deep learning model uses the RELU activation function which ensures that the model is non- linear. The COTS images are normalised to 224x224 pixels, which are input to the MobileNetV2 model, where the softmax function is used as the classifier. Activation maps, containing the features of the images, are then created. The pooling layer reduces the input size in this model and sends it to the next layer [32, 33].

## Proposed optimised CNN models

Initially the architectures of the two pre-trained models, that is, VGG19 and MobileNetV2 are used. The two deep learning models were then optimised, whereby the weights are updated at every batch to find global minima. The ADAM optimiser, which updates the learning coefficient at each batch, was applied. This technique chooses the learning rate, which is a key parameter affecting the model, based on the average first and second moment in the Root Mean Squared Propagation. The experiments conducted have shown that VGG19 outperforms MobileNetV2. To further improve the VGG19, a convolutional block attention module (CBAM) was added to the model. This module provides the identification of the important features in the COTS images. It selects the key area of the image and gears the focus of the model. CBAM is a feed-forward model whereby the starfish images are analysed from two dimensions namely channel and spatial [34]. The spatial attention is used to search the area where the attention needs to be focussed in the starfish feature map. On the other hand, the channel attention computes the internal dependency of each of the feature's channels to determine the channel that requires the attention. The channel dimension obtains its information

from the pooling layer, that is, from the average pooling and max pooling features. The spatial layer uses the output produced, that is, the concatenated feature descriptor and forwards them to the convolution layer. The classification decision is then taken to determine whether a starfish is a COTS or non-COTS. The input data X in the CBAM was first transferred into the feature map $F \in R^{C \times H \times W}$ by the former 3D CNN, where C is the number of channels, H is the height and W represents the width of the feature map before it enters the CBAM. F is then processed by the channel attention module and spatial attention module based on the following equations:

$$F' = M_c(F) \otimes F,$$

$$F'' = M_s(F) \otimes F',$$

Where, *F'* is the result of the feature map multiplying the channel map and *F"* is the result of the feature map of the spatial attention map. CNN's channel module and spatial module began to work together when CBAM added an attention function. The two modules redistributed the weight of the features in an adaptive manner after learning the essential information in both the channel dimension and the spatial dimension. As a compact and universal module, CBAM is end-to-end trainable with basic CNNs and can be readily integrated into any CNN design with minimal overhead.

## Results and discussions

To enhance the quality of the starfish images and to remove noise, the Median filter and the Gaussian filter were applied after the analysis of several enhancement algorithms. Images captured under water have more variations and are not that clear. Fig 7 shows some samples of the resulting images.

The starfish were then detected using the You Only Look Once (YOLOv4) algorithm. The latter is an object detector that has been extensively tested and achieved an optimal speed and accuracy compared to other well-known benchmarked detectors [35]. YOLOv4 is trained to analyse an image and to identify a subset of object classes. In this case, the object of interest is the starfish. Once detected, the starfish is enclosed in a bounding box. The algorithm can also detect overlapped starfishes. Fig 8 illustrates the detection of some starfishes in one of our images. Our algorithm has achieved a precision of 95%. Fig 9 shows an example where the starfishes were not detected. The clarity of the image has an effect on the object detection. In some cases, there are overlapped starfishes or complex background, where it is more challenging to detect the COTS images.

From Fig 9, a bounding box is obtained for a coral image and not for starfishes. However, in general, most of the starfishes were detected using YOLOV4 as is the case for some research work in other fields [36, 37].

### Data augmentation

Fig 10 shows the images that were generated through data augmentation. The outcome of eight (8) different data augmentation techniques on our COTS images is shown.

The custom dataset was divided into the training (70%), validation (10%) and test (20%) sets. First the architecture of VGG19 and MobileNetV2 were applied on the training set. The weights were adjusted while reducing the loss. The number of epochs were also analysed based on the loss value. The models were tested on the validation set and Fig 11 provides the initial results obtained.

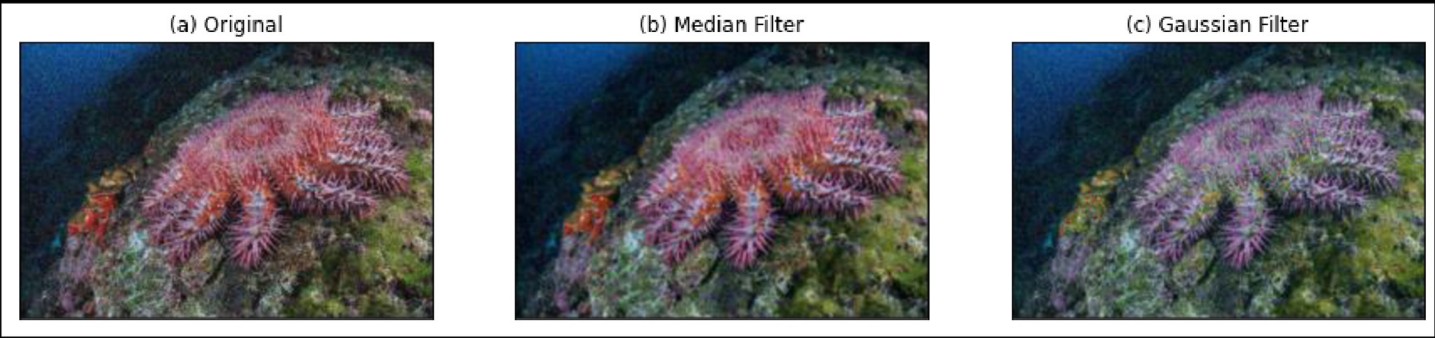

**Fig 7.** (a) Original Image (b) Image after the application of Median Filter (c) Image after the application of Gaussian filter.

From the initial results, VGG19 is overfitting and thus, it is not generalising on the validation data set. After the application of the ADAM optimiser, whereby the learning coefficient was updated at each batch, the results were improved. A training accuracy of 87.9% and a validation accuracy of 81.8% were then achieved. After an initial optimisation of the models, the latter were applied on the testing sets. A testing accuracy of 87.1% was obtained from VGG19 in contrast with MobileNetV2 where only an accuracy of 80.5% was achieved. Results obtained from a previous work in [38], also demonstrate that VGG19 outperforms MobileNetV2. It is noticed that VGG19 generalises better compared to MobileNetV2 and was thus considered for further validation. However, MobileNetV2 is more lightweight compared to VGG19, and thus consumes less power and processing time. VGG19 was thus, considered for the development of the COTS application.

Up to now, there is very little work conducted in COTS detection compared to other domains. Preliminary results show the potential of deep learning models in the detection of COTS in terms of testing accuracies. Nevertheless, these current models do not explain the intricacies of COTS image classification. To understand the input features of the deep learning models that affect the starfish classification, it is important to select the unique features that determine the decision. Thus, a convolutional block attention module was developed and

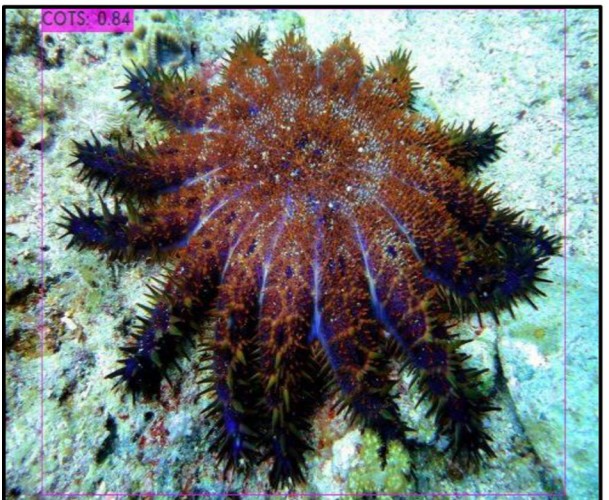 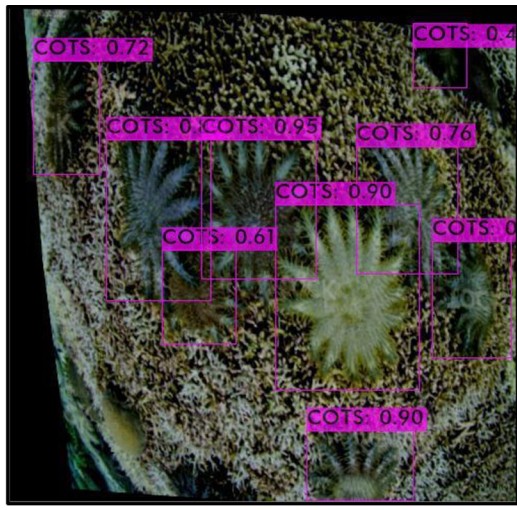

**Fig 8. Correct Detection of starfish using YOLOV4.**

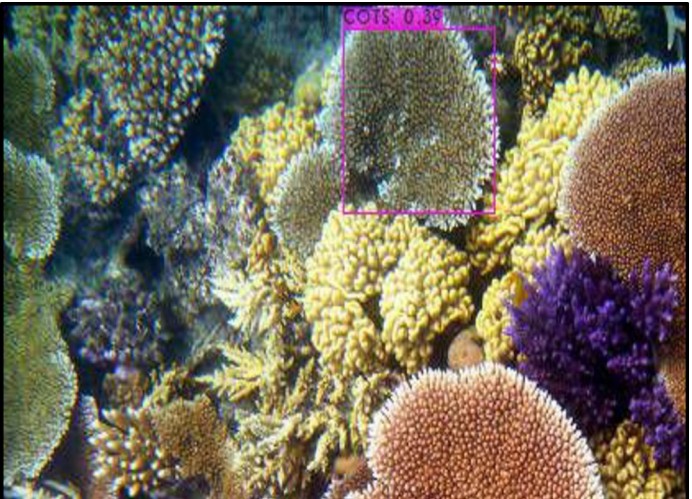

**Fig 9. Non detection of starfish.**

added to the model. Fig 12 shows some resulting images emanating from the application of the attention model.

The images highlight the determining features that contribute to the classification process. Our enhanced technique has the ability to extract the discriminative features of COTS.

To evaluate the models, other performance metrics are also considered. While accuracy is a good metric, representing the percentage of predicted COTS/ Non-COTS, we have used other metrics that provide additional information. Precision indicates the number of images that were predicted for a class and belong to that class as opposed to the total number of images predicted for that class. Likewise, Recall and F1 Score were also computed as provided in the formulae (2) and (3).

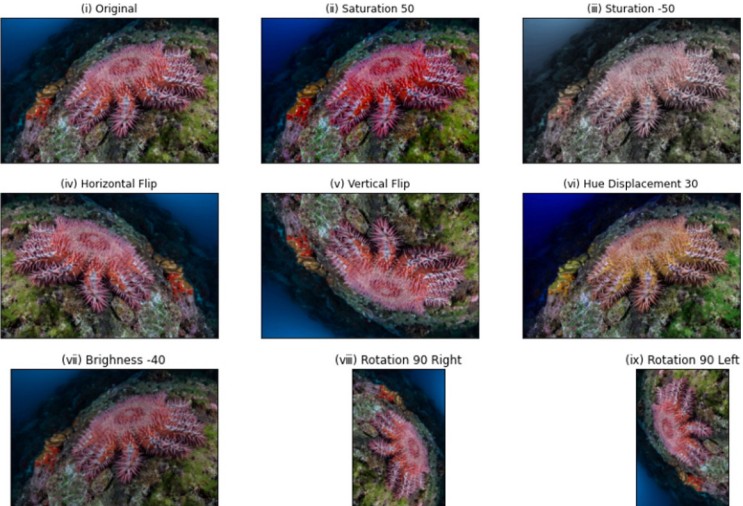

**Fig 10. Images after the application of data augmentation.**

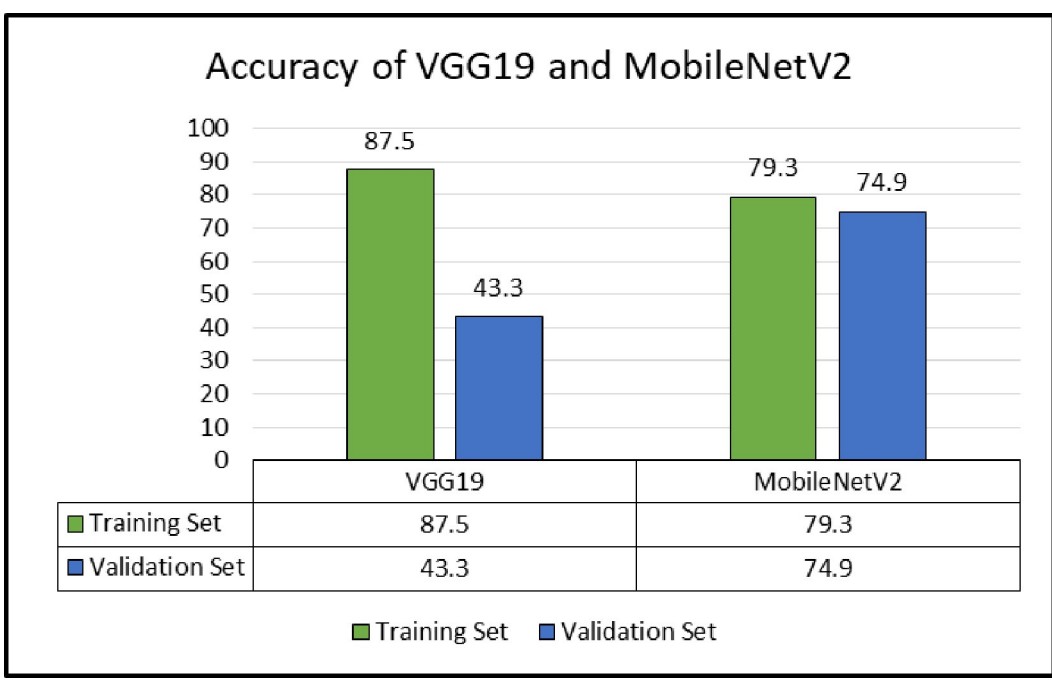

**Fig 11. Training and validation accuracies of VGG19 and MobileNetV2.**

Accuracy is the rate of correct classification and is given by the ratio between correctly classified samples and all samples.

$$Accuracy = \frac{TP + TN}{TP + TN + FP + FN} \qquad (1)$$

Where TP is the true positive, TN is the true negative

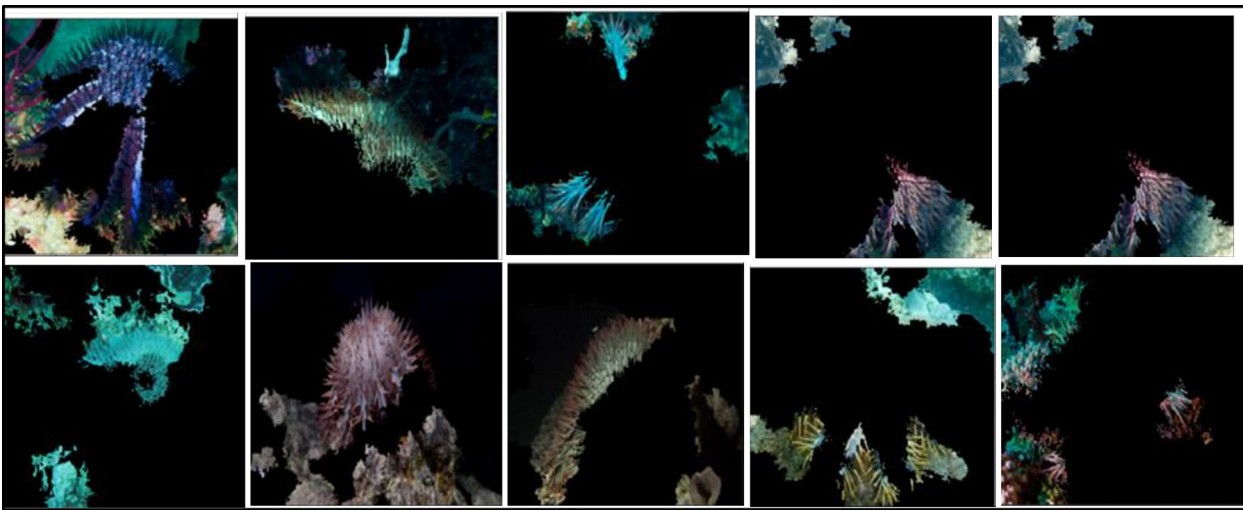

**Fig 12. Images after the application of the attention model.**

Recall is given by the ratio between correctly classified positive samples and all positive samples.

$$Recall = \frac{TP}{TP + FN} \tag{2}$$

F1- Score illustrates the harmonic mean of precision and recall.

$$F_1 - Score = 2 \times \frac{Precision \times Recall}{Precision \times Recall} = \frac{2 \times TP}{2 \times TP + FP + FN} \tag{3}$$

Table 2. shows the performance achieved by the different models.

The results indicate that an accuracy of 81.3% was attained with VGG19. The results were improved with the optimised VGG19 using ADAM optimiser. However, our D-CNN was further enhanced with the CBAM, outperforming the previous models with an accuracy of 92.6%. The precision, recall and F10 scores are also improved for the proposed enhanced VGG19 with CBAM. Lately, the hype of using the D-CNN has become intense. However, the causal features are still under-researched. Thus, there is now a growing interest in the exploration of explainable models. The latter are showing better performance in some emerging research areas. Research conducted by Aggarwal [39] (2019) showed that the attention model provides a better performance compared to a D-CNN model in the classification of skin cancer problems.

As mentioned earlier, there are only a few works that have been conducted on COTS. In one work conducted by Pooloo et al. [9], a pre-trained deep learning EfficientDet-D0 was used to detect COTS which yielded an accuracy of 81% using only 134 images for training and testing. Pooloo et al. [9] have also applied XGBoost, Decision Tree and Random Forest and have achieved an accuracy of 80.11%, 75.57% and 76.70% respectively. Our optimised model has achieved a better performance compared to existing state-of the art research work. In fact, there are very few publicly available COTS datasets, which is an impediment to the development and testing of deep learning models. In addition, up to now, no work has shown the causal parameters leading to the decision which result in a reluctance from scientists to adopt deep learning applications. CBAM determines the unique features in the images and select the ones that are more representative to COTS. From the analysis and discussions, the application of our proposed model, that is using deep learning with CBAM, is promising and can be adopted by marine scientists. The latter can explain the features that influence how the model performs classification.

## Conclusion

There is a high demand for the detection of COTS, which is destroying the coral reefs. Currently, we rely a lot on the manual detection of these starfish. Unfortunately, this technique is tedious and error-prone. Deep learning has demonstrated promising results in the field of object detection and classification. However, these techniques have not yet been explored in the detection of COTS. In this work, two well-known pre-trained deep learning models, namely VGG19 and MobileNetv2, have been applied to a custom COTS dataset, where images were locally collected in the Mauritian lagoon and taken in online datasets as well. The weights and the hyperparameters have been fine-tuned to reach an optimal solution. Data augmentation was also applied on the datasets. VGG19 outperformed MobileNetv2 in the detection of COTS. Though deep learning has the capability of detecting objects, the features that determine the decision are not known. While the accuracy does give an idea of the number of images that have been predicted correctly, it is not capable of providing sufficient information.

**Table 2. Models performance for VGG19.**

| Model | Accuracy | Precision | Recall | F1-Score |
|---|---|---|---|---|
| **VGG19** | 81.3 | 0.86 | 0.75 | 0.80 |
| **Optimised VGG19 with ADAM optimiser** | 87.1 | 0.93 | 0.80 | 0.85 |
| **Enhanced VGG19 with CBAM (Proposed Model)** | 92.6 | 0.95 | 0.92 | 0.92 |
| **Pooloo et al. [9]–** | | - | - | - |
| **1. XGBoost** | 1. 80.11 | | | |
| **2. Decision Trees** | 2. 75.57 | | | |
| **3. Random Forest** | 3. 76.70 | | | |
| **Pooloo et al., [9] EfficientDet-D0 (Only detection of COTS and not classification)** | 81 | - | - | - |

To further improve the model and to show the causal parameters, an attention model was used. The latter shows the features that influence the decision, that is, the detection of COTS or non-COTS. The resulting work can be used in oceanographic research institutions by the government and non-governmental organisations to protect reef habitats. It is known that the quality of images captured underwater can be compromised. There are scopes for further exploration to improve the quality of the images, which eventually has an impact on the performance at later stages. The proposed system can be used in real-time for the detection and classification of COTS in the lagoons to help detect and manage COTS outbreaks.

## Author Contributions

**Conceptualization:** Maleika Heenaye- Mamode Khan, Anjana Makoonlall.

**Data curation:** Nadeem Nazurally.

**Formal analysis:** Maleika Heenaye- Mamode Khan, Anjana Makoonlall, Zahra Mungloo-Dilmohamud.

**Investigation:** Maleika Heenaye- Mamode Khan, Anjana Makoonlall, Nadeem Nazurally.

**Methodology:** Anjana Makoonlall.

**Project administration:** Maleika Heenaye- Mamode Khan.

**Resources:** Maleika Heenaye- Mamode Khan, Nadeem Nazurally.

**Software:** Anjana Makoonlall.

**Supervision:** Maleika Heenaye- Mamode Khan.

**Validation:** Maleika Heenaye- Mamode Khan, Zahra Mungloo- Dilmohamud.

**Visualization:** Maleika Heenaye- Mamode Khan, Anjana Makoonlall, Zahra Mungloo-Dilmohamud.

**Writing – original draft:** Maleika Heenaye- Mamode Khan, Anjana Makoonlall, Nadeem Nazurally, Zahra Mungloo- Dilmohamud.

**Writing – review & editing:** Maleika Heenaye- Mamode Khan, Anjana Makoonlall, Nadeem Nazurally, Zahra Mungloo- Dilmohamud.

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
