## [Decision Letter · Decision Letter 0]

18 Oct 2022

PONE-D-22-23986Attention- based Convolutional Neural Network (CNN) for the Identification of Crown of Thorns Starfish (COTS)PLOS ONE

Dear Dr. Heenaye- Mamode Khan,

Thank you for submitting your manuscript to PLOS ONE. After careful consideration, we feel that it has merit but does not fully meet PLOS ONE’s publication criteria as it currently stands. Therefore, we invite you to submit a revised version of the manuscript that addresses the points raised during the review process.

We look forward to receiving your revised manuscript.

Kind regards,

Kapil Kumar Nagwanshi, PhD

Academic Editor

PLOS ONE

Journal Requirements:

3. Please ensure that you include a title page within your main document. We do appreciate that you have a title page document uploaded as a separate file, however, as per our author guidelines (http://journals.plos.org/plosone/s/submission-guidelines#loc-title-page) we do require this to be part of the manuscript file itself and not uploaded separately.

"NO authors have competing interests"

Reviewers' comments:

Reviewer's Responses to Questions

**Comments to the Author**

1. Is the manuscript technically sound, and do the data support the conclusions?

Reviewer #1: Yes

Reviewer #2: Partly

Reviewer #3: Partly

2. Has the statistical analysis been performed appropriately and rigorously? 

Reviewer #1: Yes

Reviewer #2: No

Reviewer #3: No

3. Have the authors made all data underlying the findings in their manuscript fully available?

Reviewer #1: Yes

Reviewer #2: Yes

Reviewer #3: Yes

4. Is the manuscript presented in an intelligible fashion and written in standard English?

Reviewer #1: Yes

Reviewer #2: Yes

Reviewer #3: Yes

5. Review Comments to the Author

Reviewer #1: The authors needs to update their research article as per the below mentioned comments:

1. The difficulties of snorkelling and diving must be discussed by the author in order to identify COTS.

2. The author should update Fig. 2. (Improve the readability of Fig. 2)

3. Modify Fig. 3, Fig. 6. Difficult to read

4. Change Fig. 9 (Captions of Fig. 9(a) to Fig. 9(i) are not clearly visible)

6. Authors must include the full forms of YOLO4 and VGG19 in the Contribution section.

7. The authors have used VGG19 with CBM, which performs better than earlier models. The characteristics of the convolutional block attention module (CBAM) must be described by authors.

Reviewer #2: The following points need to be looked upon in the manuscript.

1. The review of literature provides an extremely low number of papers in Table 1. Are these papers important ones for the readers with respect to the topic of research, or only these many number of papers are available in this field?

2. The datasets links must be changed to the citations and provide the links as part of reference.

3. The figures are not placed in the proper places in the manuscript.

4. The extraction of features using VGG16 and MobileNetV2 are already proposed ones. The manuscript must highlight the importance of using these techniques in the proposal. Also, justify their usage in comparison with other techniques available in this domain.

5. Table 3 indicates the performance comparison. Also provide a written explanation on why these values are obtained as such and reason of such comparative values. Justify the proposed model's uniqueness in being better than the other approaches.

6. The results are not adequate and lacks the amount of work done to justify the proposal.

7. The discussion on future scope of work must be clearly explained.

Reviewer #3: Review

Abstract: - Is there any improvement of map (mean average precision), Loss and FPS of proposed model (attention based with CNN) over VGG-19 and MobileNetV2 model? Please give improvement results in percentage.

Dataset:- Number of images in data sets are not enough large for approximate results. Is there any large size of data set available?

No any figures are available or visible like fig 1,2,3,4……etc in that manuscript. Kindly provide the all figures.

Proposed Model:- In that manuscript ADAM used as optimizer along with VGG-19 and MobileNetV2 . How did you optimize your attention based model (CBAM) for further enhancement of VGG19? Moreover there is no any working process and math behind the reason about proposed model (spatial attention of CBAM) so that it could be understood that how attention model focused only on the particular area of COTS even how could fed the CBAM along with CNN variant’s layer as so called optimizer/enhancement. Kindly provide the working process and math of the proposed model.

Results and Discussion:- In that manuscript, in results section,YoLOV4 version used to detect the starfish but why did not compare YOLOV4 results with your proposed model and other VGG19 and MobileNetV2? Wrote there in manuscript “Our algorithm has achieved an accuracy of 95%”, is that YOLOV4 accuracy? Or your proposed model accuracy? There is lot confusion.

Is YOLOV4’s result better than of your proposed model (CBAM)? Kindly provide the results of YOLO V4 and compare it with proposed model along their graphs.

In figure 10, there should have testing accuracy and loss results of all variant too. Kindly provide.

6. PLOS authors have the option to publish the peer review history of their article (what does this mean?). If published, this will include your full peer review and any attached files.

Reviewer #1: No

Reviewer #2: No

Reviewer #3: **Yes: **Manjit Jaiswal

---

## [Author Response · Author response to Decision Letter 0]

6 Dec 2022

Response to Reviewer

Journal Requirements:

File naming revised

Codes are available upon request and may be uploaded on GitHub if required.

3. Please ensure that you include a title page within your main document. We do appreciate that you have a title page document uploaded as a separate file, however, as per our author guidelines (http://journals.plos.org/plosone/s/submission-guidelines#loc-title-page) we do require this to be part of the manuscript file itself and not uploaded separately.

Title page has been included

"NO authors have competing interests"

This has been added in the cover letter. 

We have added the links in the cover letter 

Response to the reviewer’s Comments

Reviewer #1: The authors needs to update their research article as per the below mentioned comments:

1. The difficulties of snorkelling and diving must be discussed by the author in order to identify COTS.

 The difficulties of snorkelling and diving was mentioned in the abstract.

We have added the following the the abstract: Where strong currents result in poor image capture, damage of capturing equipment, and are of high risks.

2. The author should update Fig. 2. (Improve the readability of Fig. 2)

 This has been updated to a higher resolution 

3. Modify Fig. 3, Fig. 6. Difficult to read

 This has been updated to a higher resolution

4. Change Fig. 9 (Captions of Fig. 9(a) to Fig. 9(i) are not clearly visible)

 This has been updated to a higher resolution

6. Authors must include the full forms of YOLO4 and VGG19 in the Contribution section.

 We have put the full form of YOLOV4 and VGG19 in the contribution section 

7. The authors have used VGG19 with CBM, which performs better than earlier models. The characteristics of the convolutional block attention module (CBAM) must be described by authors Section 3.5 has been updated.

Reviewer #2: The following points need to be looked upon in the manuscript.

1. The review of literature provides an extremely low number of papers in Table 1. Are these papers important ones for the readers with respect to the topic of research, or only these many number of papers are available in this field?

 These are the only papers in this field at the time of submission. Only these authors have worked in the area of COTs and deep learning 

2. The datasets links must be changed to the citations and provide the links as part of reference. The manuscript has been updated accordingly. 

3. The figures are not placed in the proper places in the manuscript. Figures have been referred by the caption. Figures have been placed so as not to disrupt the format of the manuscript and have been placed following their mention in the paper. However, since images are uploaded separately from the main text, and the final Pdf is generated automatically, the generated Pdf may be different from the intended manuscript. 

4. The extraction of features using VGG16 and MobileNetV2 are already proposed ones. The manuscript must highlight the importance of using these techniques in the proposal. Also, justify their usage in comparison with other techniques available in this domain. It is mentioned in Section 3.4 that : 

Despite the promising results of deep learning, the latter has not yet been explored in the detection of COTS

We have added the following to address this comment: 

Instead of using a single Convolution layer with a big kernel size, the VGG network introduced the idea of grouping many convolution layers with smaller kernel sizes. As a result, there were fewer features at the output, and adding three RELU layers rather than one increased the number of learning instances. On the other hand, MobileNet-v2 is a convolutional neural network that is 53 layers deep, which has been trained with over one million images from the ImageNet dataset. 

We have also highlighted the importance of these techniques over machine learning techniques. 

We have added the following:

Previously machine learning techniques were popular in feature extraction and selection. Machine learning refers to the use of algorithms by computers to learn from data and carry out tasks automatically without explicit programming. In contrast, deep learning employs a powerful set of algorithms that are designed after the human brain. This makes it possible to process unstructured data, including text, photos, and documents. In this case, deep learning is able to extract complex features from the images.

5. Table 3 indicates the performance comparison. Also provide a written explanation on why these values are obtained as such and reason of such comparative values. Justify the proposed model's uniqueness in being better than the other approaches.

 Compare it with results in machine learning – COTS

6. The results are not adequate and lacks the amount of work done to justify the proposal. The section Results and Discussions has been amended. 

7. The discussion on future scope of work must be clearly explained.

 The following has been added in the conclusion section:

The proposed system can be used in real-time for the detection and classification of COTS in the lagoons to help detect and manage COTS outbreaks.

Reviewer #3: Review

Abstract: - Is there any improvement of map (mean average precision), Loss and FPS of proposed model (attention based with CNN) over VGG-19 and MobileNetV2 model? Please give improvement results in percentage.

This has been taken care of in the abstract. 

Dataset:- Number of images in data sets are not enough large for approximate results. Is there any large size of data set available?

The Kaggle dataset was the largest publicly available dataset. In addition, data augmentation techniques were applied to increase the number of images in the dataset.

No any figures are available or visible like fig 1,2,3,4……etc in that manuscript. Kindly provide the all figures.

Images are uploaded separately from the main text, and the final Pdf is generated automatically, the generated Pdf may be different from the intended manuscript. Image quality has been further enhanced and re-submitted. 

Proposed Model:- In that manuscript ADAM used as optimizer along with VGG-19 and MobileNetV2 . How did you optimize your attention based model (CBAM) for further enhancement of VGG19? Moreover there is no any working process and math behind the reason about proposed model (spatial attention of CBAM) so that it could be understood that how attention model focused only on the particular area of COTS even how could fed the CBAM along with CNN variant’s layer as so called optimizer/enhancement. Kindly provide the working process and math of the proposed model.

More details of the proposed model have been added with some Mathematical equations. 

Results and Discussion: - In that manuscript, in results section,YoLOV4 version used to detect the starfish but why did not compare YOLOV4 results with your proposed model and other VGG19 and MobileNetV2? Wrote there in manuscript “Our algorithm has achieved an accuracy of 95%”, is that YOLOV4 accuracy? Or your proposed model accuracy? There is lot confusion.

95% was for the precision. The text has been amended accordingly. 

Is YOLOV4’s result better than of your proposed model (CBAM)? Kindly provide the results of YOLO V4 and compare it with proposed model along their graphs.

In figure 10, there should have testing accuracy and loss results of all variant too. Kindly provide.

Please note that YOLO has been used for detection and CBAM for classification and Explainability 

6. PLOS authors have the option to publish the peer review history of their article (what does this mean?). If published, this will include your full peer review and any attached files.

Do you want your identity to be public for this peer review? For information about this choice, including consent withdrawal, please see our Privacy Policy.

Reviewer #1: No

Reviewer #2: No

Reviewer #3: Yes: Manjit Jaiswal

---

## [Decision Letter · Decision Letter 1]

3 Mar 2023

Identification of Crown of Thorns Starfish (COTS) using Convolutional Neural Network (CNN) and Attention Model

PONE-D-22-23986R1

Dear Dr. Heenaye- Mamode Khan,

We’re pleased to inform you that your manuscript has been judged scientifically suitable for publication and will be formally accepted for publication once it meets all outstanding technical requirements.

Kind regards,

Kapil Kumar Nagwanshi, PhD

Academic Editor

PLOS ONE

Additional Editor Comments (optional):

Reviewers' comments:

Reviewer's Responses to Questions

**Comments to the Author**

1. If the authors have adequately addressed your comments raised in a previous round of review and you feel that this manuscript is now acceptable for publication, you may indicate that here to bypass the “Comments to the Author” section, enter your conflict of interest statement in the “Confidential to Editor” section, and submit your "Accept" recommendation.

Reviewer #1: All comments have been addressed

Reviewer #4: All comments have been addressed

2. Is the manuscript technically sound, and do the data support the conclusions?

Reviewer #1: Yes

Reviewer #4: Yes

3. Has the statistical analysis been performed appropriately and rigorously? 

Reviewer #1: Yes

Reviewer #4: Yes

4. Have the authors made all data underlying the findings in their manuscript fully available?

Reviewer #1: Yes

Reviewer #4: Yes

5. Is the manuscript presented in an intelligible fashion and written in standard English?

Reviewer #1: Yes

Reviewer #4: Yes

6. Review Comments to the Author

Reviewer #1: The authors have addressed all the comments. No further changes are required by the authors. The manuscript is finally accepted without any revision.

Reviewer #4: Author/s tried to address the comments of reviewers but could not do it properly. Here are few more comments for the submitted manuscript.

1. In table-1, row 1 and row 2, references are strike.

2. Figure 2, still need improvement with good quality.

3. 33 reference was not cited.

Some important references related to ML, DL, AI are missing. Author/s, Please add recent papers published in last three years for Application of AI, ML and DL. Few good papers which are published within last 3 years on AI, ML and DL are as below. Include such papers in related work section with comparative study.

-https://doi.org/10.1007/s11831-022-09738-3

-https://doi.org/10.1007/s11831-022-09816-6

-https://doi.org/10.1007/s11042-022-12891-3

-https://doi.org/10.1007/s13042-022-01591-x

7. PLOS authors have the option to publish the peer review history of their article (what does this mean?). If published, this will include your full peer review and any attached files.

Reviewer #1: No

Reviewer #4: No

---

## [Editor Report · Acceptance letter]

15 Mar 2023

PONE-D-22-23986R1 

Identification of Crown of Thorns Starfish (COTS) using Convolutional Neural Network (CNN) and Attention Model 

Dear Dr. Heenaye- Mamode Khan:

I'm pleased to inform you that your manuscript has been deemed suitable for publication in PLOS ONE. Congratulations! Your manuscript is now with our production department. 

Kind regards, 

on behalf of

Dr. Kapil Kumar Nagwanshi 

Academic Editor

PLOS ONE